# A Proposed Personalized Spine Care Protocol (SpineScreen) to Treat Visualized Pain Generators: An Illustrative Study Comparing Clinical Outcomes and Postoperative Reoperations between Targeted Endoscopic Lumbar Decompression Surgery, Minimally Invasive TLIF and Open Laminectomy

**DOI:** 10.3390/jpm12071065

**Published:** 2022-06-29

**Authors:** Kai-Uwe Lewandrowski, Ivo Abraham, Jorge Felipe Ramírez León, Albert E. Telfeian, Morgan P. Lorio, Stefan Hellinger, Martin Knight, Paulo Sérgio Teixeira De Carvalho, Max Rogério Freitas Ramos, Álvaro Dowling, Manuel Rodriguez Garcia, Fauziyya Muhammad, Namath Hussain, Vicky Yamamoto, Babak Kateb, Anthony Yeung

**Affiliations:** 1Fundación Universitaria Sanitas, Clínica Reina Sofía-Clínica Colsanitas, Centro de Columna-Cirugía Mínima Invasiva, Bogotá 104-76, D.C., Colombia; 2The Federal University of the State of Rio de Janeiro UNIRIO, Pain and Spine Minimally Invasive Surgery Service at Gaffrée Guinle University Hospital HUGG, Tijuca, Rio de Janeiro 20270-004 RJ, Brazil; 3Center for Advanced Spine Care of Southern Arizona and Surgical Institute of Tucson, Tucson, AZ 85712, USA; 4Pharmacy Practice and Science, Family and Community Medicine, Clinical Translational Sciences at the University of Arizona, Roy P. Drachman Hall, Rm. B306H, Tucson, AZ 85721, USA; abraham@pharmacy.arizona.edu; 5Minimally Invasive Spine Center Bogotá D.C. Colombia, Reina Sofía Clinic Bogotá D.C. Colombia, Department of Orthopaedics Fundación Universitaria Sanitas, Bogotá 104-76, D.C., Colombia; jframirezl@yahoo.com; 6Department of Neurosurgery, Rhode Island Hospital, The Warren Alpert Medical School of Brown University, Providence, RI 12321, USA; atelfeian@Lifespan.org; 7Advanced Orthopedics, 499 East Central Parkway, Altamonte Springs, FL 32701, USA; mloriomd@gmail.com; 8Department of Orthopedic Surgery, Arabellaklinik, 81925 Munich, Germany; hellinger@gmx.de; 9The Weymouth Hospital, 42-46 Weymouth Street London, 27 Harley Street, London W1G 9QP, UK; mknight@spinal-foundation.org; 10Pain and Spine Minimally Invasive Surgery Service at Gaffre e Guinle University Hospital, Rio de Janeiro 20270-004 RJ, Brazil; paulo.carvalho@unirio.br; 11Orthopedic Clinics at Gaffrée Guinle University Hospital HUGG, Rio de Janeiro 20270-004 RJ, Brazil; drmaxrmos@hotmail.com; 12Orthopaedic Spine Surgeon, Director of Endoscopic Spine Clinic, Santiago 8330024, Chile; adowling@dws.cl; 13Department of Orthopaedic Surgery, USP, Ribeirão Preto 14049-900 SP, Brazil; 14Spine Clinic, The American-Bitish Cowdray Medical Center I.A.P. Campus Santa Fe, México City 87501, Mexico; drmrodriguez@yahoo.com.mx; 15Society for Brain Mapping and Therapeutics (SBMT), Los Angeles, CA 90272, USA; fauziyya-muhammad@ouhsc.edu (F.M.); nhussain@llu.edu (N.H.); vyamamot@usc.edu (V.Y.); babak.kateb@worldbrainmapping.org (B.K.); 16Brain Mapping Foundation (BMF), Los Angeles, CA 90272, USA; 17Department of Neurosurgery, Loma Linda University, Loma Linda, CA 90272, USA; 18USC-Norris Comprehensive Cancer Center, USC-Keck School of Medicine, Los Angeles, CA 90033, USA; 19Middle East Brain + Initiative, Los Angeles, CA 90272, USA; 20National Center for Nanobioelectronics, Los Angeles, CA 90272, USA; 21Desert Institute for Spine Care, Phoenix, AZ 85058, USA; ayeung@sciatica.com

**Keywords:** pain generators, lumbar decompression surgery, lumbar foraminal and lateral recess stenosis, durability, postoperative natural history, reoperation, aftercare

## Abstract

Background: Endoscopically visualized spine surgery has become an essential tool that aids in identifying and treating anatomical spine pathologies that are not well demonstrated by traditional advanced imaging, including MRI. These pathologies may be visualized during endoscopic lumbar decompression (ELD) and categorized into primary pain generators (PPG). Identifying these PPGs provides crucial information for a successful outcome with ELD and forms the basis for our proposed personalized spine care protocol (SpineScreen). Methods: a prospective study of 412 patients from 7 endoscopic practices consisting of 207 (50.2%) males and 205 (49.8%) females with an average age of 63.67 years and an average follow-up of 69.27 months was performed to compare the durability of targeted ELD based on validated primary pain generators versus image-based open lumbar laminectomy, and minimally invasive lumbar transforaminal interbody fusion (TLIF) using Kaplan-Meier median survival calculations. The serial time was determined as the interval between index surgery and when patients were censored for additional interventional and surgical treatments for low back-related symptoms. A control group was recruited from patients referred for a surgical consultation but declined interventional and surgical treatment and continued on medical care. Control group patients were censored when they crossed over into any surgical or interventional treatment group. Results: of the 412 study patients, 206 underwent ELD (50.0%), 61 laminectomy (14.8%), and 78 (18.9%) TLIF. There were 67 patients in the control group (16.3% of 412 patients). The most common surgical levels were L4/5 (41.3%), L5/S1 (25.0%), and L4-S1 (16.3%). At two-year f/u, excellent and good Macnab outcomes were reported by 346 of the 412 study patients (84.0%). The VAS leg pain score reduction was 4.250 ± 1.691 (*p* < 0.001). No other treatment during the available follow-up was required in 60.7% (125/206) of the ELD, 39.9% (31/78) of the TLIF, and 19.7% (12/61 of the laminectomy patients. In control patients, only 15 of the 67 (22.4%) control patients continued with conservative care until final follow-up, all of which had fair and poor functional Macnab outcomes. In patients with Excellent Macnab outcomes, the median durability was 62 months in ELD, 43 in TLIF, and 31 months in laminectomy patients (*p* < 0.001). The overall survival time in control patients was eight months with a standard error of 0.942, a lower boundary of 6.154, and an upper boundary of 9.846 months. In patients with excellent Macnab outcomes, the median durability was 62 months in ELD, 43 in TLIF, and 31 months in laminectomy patients versus control patients at seven months (*p* < 0.001). The most common new-onset symptom for censoring was dysesthesia ELD (9.4%; 20/206), axial back pain in TLIF (25.6%;20/78), and recurrent pain in laminectomy (65.6%; 40/61) patients (*p* < 0.001). Transforaminal epidural steroid injections were tried in 11.7% (24/206) of ELD, 23.1% (18/78) of TLIF, and 36.1% (22/61) of the laminectomy patients. The secondary fusion rate among ELD patients was 8.8% (18/206). Among TLIF patients, the most common additional treatments were revision fusion (19.2%; 15/78) and multilevel rhizotomy (10.3%; 8/78). Common follow-up procedures in laminectomy patients included revision laminectomy (16.4%; 10/61), revision ELD (11.5%; 7/61), and multilevel rhizotomy (11.5%; 7/61). Control patients crossed over into ELD (13.4%), TLIF (13.4%), laminectomy (10.4%) and interventional treatment (40.3%) arms at high rates. Most control patients treated with spinal injections (55.5%) had excellent and good functional outcomes versus 40.7% with fair and poor (3.7%), respectively. The control patients (93.3%) who remained in medical management without surgery or interventional care (14/67) had the worst functional outcomes and were rated as fair and poor. Conclusions: clinical outcomes were more favorable with lumbar surgeries than with non-surgical control groups. Of the control patients, the crossover rate into interventional and surgical care was 40.3% and 37.2%, respectively. There are longer symptom-free intervals after targeted ELD than with TLIF or laminectomy. Additional intervention and surgical treatments are more often needed to manage new-onset postoperative symptoms in TLIF- and laminectomy compared to ELD patients. Few ELD patients will require fusion in the future. Considering the rising cost of surgical spine care, we offer SpineScreen as a simplified and less costly alternative to traditional image-based care models by focusing on primary pain generators rather than image-based criteria derived from the preoperative lumbar MRI scan.

## 1. Introduction

Modern spine care plays out in cash-strapped health care systems grappling with improving patient outcomes through value-driven treatments while minimizing risks and controlling costs. Patient-centered, cost-effective, financially sustainable spine care models are needed to improve clinical outcomes and avoid unnecessary surgery. Personalized patient-focused care models may potentially be at odds with traditional spinal surgery protocols that are founded on image-based population management strategies [1,2,3] and often demand failed-non-operative-care before meeting medical necessity criteria to warrant authorization for intervention or spine surgery to relieve pain related to neural element compression, deformity, and instability. Examples include coverage guidelines by the Center for Medicare Services (CMS) and health insurance companies requiring more than 4 mm of mobile anterior-posterior spondylolisthesis for lumbar decompression fusion approval [4,5,6,7,8,9,10,11,12,13,14,15,16,17]. Other perhaps even more controversial criteria are centered around the perceived severity of the neural element compression [18]. Only the most severe cases of spinal stenosis in the central or lateral canal are deemed appropriate for surgery [19,20,21,22,23,24,25]. Often patients are told that they are too young or old or their condition is not bad enough to justify surgical treatment. Consequently, patients who remain in pain create a significant unmet demand for cost-effective alternative non-surgical care, which is not necessarily lower in cost.

Institutionalized medical practices may prefer to embrace population-based management strategies as they are more suitable for directing patients through a complex screening process. The patient flow is often managed by support staff, who sometimes have more input into identifying the most appropriate next point of care by checking patients for the presence of advanced disease eligible for medical, interventional, or surgical treatment [26,27]. The result is a costly and labor-intensive bureaucratic process with many repetitive and often ineffective rounds of referrals to non-operative subspecialists before considering a consultation with an orthopedic or neurological spinal surgeon [28]. At that point, the definitive care in many cases is aggressive as the disease, by design, was allowed to progress to its end-stage, where costly spinal fusion is often the only option. Attempting to roll spine care into one main treatment episode, similar to total knee and hip replacements for end-stage osteoarthritis, is another commonly employed yet expensive strategy many patients do not consider [28]. Others get lost in the referral maze and return to their referring primary care physicians for medical management. At least in part, these repetitive and ineffective referrals may contribute to the ongoing opioid epidemic. Some patients manage the problem by turning to other less scrutinized alternative cash-based treatments. However, these may not necessarily be less costly either. The additional societal burden created by the loss of work hours and mounting disability is staggering [29,30,31,32,33,34,35,36,37,38,39,40,41,42].

Early and staged treatment of painful lumbar spine disease is rarely considered [28]. Patients are reassured about the benign natural history of a herniated disc and spinal stenosis by proclaiming that symptoms should soon resolve with non-steroidal anti-inflammatories, physical therapy, and activity modification with short-term bed rest and early return to work whenever possible [43,44,45,46]. Nonetheless, many of these patients suffer from painful inflammatory or compressive conditions affecting the neural elements due to leakage from toxic annular tears and irritated facet joint cysts [47,48,49,50,51]. Many other pain generators frequently escape the routine lumbar MRI scan. These include a herniated or inflamed disc, an inflamed nerve, a hypervascular scar, a hypertrophied superior articular process (SAP) and ligamentum flavum, a tender capsule, an impacting facet margin, a superior foraminal facet osteophyte, a superior foraminal ligament impingement, a hidden shoulder osteophyte, a tethered and multiple furcal nerve roots, contracted foraminal ligaments, intra-annular granulation tissue, delaminated and fissured disc tissue, and many others [Figure 1]. [1,28,50,52] Patients in this traditional spine care watershed area are virtually unattended without meaningful remedies and left by default to homeopathic, naturopathic, chiropractic, and interventional pain management care because they cannot get any help anywhere else [52]. More recently, regenerative medicine treatment strategies are emerging with platelet-enriched plasma preparations (PRP) and stem cells injected into the diseased intervertebral lumbar disc or facet joints to promote healing and pain relief [53,54,55]. However, their effectiveness and safety record are yet to be thoroughly investigated.

A personalized and more targeted approach to spine care with a focus on early treatment of validated primary pain generators (PPG) would not only improve accuracy but perhaps decrease overutilization of services and reduce costly surgical aftercare that is prompted by the natural history of the index operation which is many cases is an accelerated structural decline requiring more instrumented fusion surgery. The authors took an approach similar to the NeuroScreen method proposed by the Society for Brain Mapping and Therapeutics (SBMT) by describing their multi-faceted methodology (SpineScreen) to find the pain generators responsible for the patient’s disability. Furthermore, they stipulated that this methodology could form the basis for a more patient-centric, financially sustainable care model by not letting the disease run astray without management to its end-stage and alleviating patient and clinician frustration with the current system where it is tough to get definitive help early on. Minimally invasive and endoscopic lumbar decompression surgery (ELD), in particular, can fill the gap in the current care models created by the unmet patient demand.

An increasing number of surgeons recognize its ability to treat validated spinal pain generators ahead of irreversible structural damage to allow the human spine to heal without creating iatrogenic and approach-related problems such as scar tissue [56,57] and instability [58,59,60,61]. In this study, the authors investigated clinical outcomes with a patient-centered personalized approach to spine care that essentially ignores those traditional image-based necessity criteria for surgery focused on deformity and instability but employed a screening protocol to identify the painful pathology. The authors offer a comparative durability analysis of symptom-free intervals after ELD, minimally invasive transforaminal lumbar interbody fusion (TLIF), and open laminectomy in comparison to control group patients who declined surgical and interventional treatment to illustrate the diverse postoperative natural history displayed by patients who had these three different types of lumbar spine surgeries.

## 2. Materials and Methods

### 2.1. Patient Population

This prospective observational cohort study included 412 patients pooled from 7 endoscopic practices and supporters of the personalized endoscopic spine concept who underwent either traditional open laminectomy (61 patients), minimally invasive TLIF (78 patients), or outpatient minimally invasive targeted ELD (206 patients) for symptomatic lumbar herniated disc, toxic annular tears, or spinal stenosis non-responsive to a minimum of 6 weeks of medical and interventional spine care. A control group was recruited from 67 patients referred for a surgical consultation but declined interventional and surgical treatment and continued on medical care. Specifically, patients with long-term follow-up were sought who had their surgeries between 2000 and 2016 to evaluate the postoperative natural history after the index surgery versus control group patients who were also referred for a surgical consultation but declined surgery or interventional pain management. Patients were followed for a mean of 69.27 months, ranging from 39 to 118 months, with a standard deviation of 12.69 months. The patients’ ages ranged from 26 to 88 years, with a mean age of 63.67. Q-Q plot analysis showed normal age distribution among study patients (Figure 2 and Figure 3).

There were 205 (50.2%) male and 207 (49.8%) female patients. Patients were enrolled in this study employing inclusion and exclusion criteria published elsewhere. (22–25) In brief, all patients were treated for bony and soft tissue stenosis affecting the central canal, lateral recess, and neuroforamina. Foraminal and lateral recess stenosis was treated with ELD. Patients with severe central canal stenosis were treated with laminectomy. Patients with mobile (>4 mm) anterolisthesis on dynamic extension/flexion views were considered for TLIF.

### 2.2. Preoperative Work Up and Surgical Decision Making

Patients were thoroughly interviewed to obtain an accurate history of their complaints to arrive at a personalized care plan [26]. At the initial consultation, walking distance and the overall endurance limit were recorded to distinguish acute sciatica-type- from neurogenic claudication symptoms. A thorough physical examination and review of the pertinent lumbar plain film studies, including at a minimum posteroanterior (PA), lateral (LAT), extension/flexion views, and advanced imaging studies, including MRI and CT scans, were recorded. Diagnostic injections were performed at the suspected symptomatic level with 1% lidocaine using previously vetted and published protocols [62,63,64,65,66,67]. If a patient-reported pain reduction on the VAS scale > 50%, the injection was considered diagnostic and predictive of a favorable outcome with surgical decompression [67]. This protocol was used in conjunction with matching clinical symptoms and sup-porting history and physical examination to identify a single level as the predominant pain generator regardless of the MRI findings at that level [1]. Several radiographic classifications [3,21,22,23,67,68,69] of foraminal and lateral recess stenosis described elsewhere were employed by the authors to record and grade the preoperative MRI scan by defining the location and extent of the offending painful pathology – the primary pain generator [21,70,71] if it could be identified. In some cases, patients would undergo surgery with a negative MRI scan recognizing that up to 35% of patients referred for sciatica-type low back and leg pain may be false negative on routine radiological grading of the preoperative MRI scan [1]. The authors present this combination of diagnostic steps during the workup of patients with unrelenting low back pain-related symptoms as the SpineScreen methodology.

### 2.3. Clinical Follow-Up

The clinical success of the personalized surgical care plan was assessed as reductions in the VAS for leg pain ranging from no pain (0) to worst pain (10) [72] and the Macnab criteria [73] being used as the primary outcome measures to assess the functional improvement at two-years from the index operation. After that, patients were asked at each available follow-up visit whether they had any deviation from an uneventful postoperative course requiring spinal injections or reoperations. More than five-year follow-up data were available in 80.58% (332/412) of study patients.

### 2.4. Surgical Techniques and Postoperative Rehabilitation

The targeted ELD was achieved via the endoscopic transforaminal approach [47,48,51,74,75,76,77,78,79]. Patients underwent a foraminoplasty via partial resection of the superior articular process and the caudal pedicle [51]. Additionally, bony and soft tissue stenosis in the lateral spinal canal was treated. The painful pathology could be directly visualized on the magnified video screen during the irrigated endoscopic decompression surgery. The primary pain generator (PPG) was recorded for each ELD patient and cross-tabulated with the MRI scan report in a dichotomized manner as either positive or negative. An exemplary case is shown in Figure 4.

Laminectomy patients underwent a traditional open midline incision with subperiosteal dissection of the paraspinal muscles (Figure 4g) [72]. TLIF patients underwent minimally invasive fusion surgery using widely accepted surgical techniques employing a tubular retractor system, four pedicle screws, and at least one interbody fusion cage (Figure 4h) [70,71,72,73,74,75,76,77,78,79,80,81,82,83,84,85]. Pain generators could generally not be directly visualized reliably due to bleeding and lack of constant irrigation. Most patients mobilized rapidly and did not require postoperative rehabilitation. Postsurgical dorsal root ganglion inflammation caused dysesthesia in some patients [86]. If present, it was treated with non-steroidal anti-inflammatories, gabapentin or pregabalin, and transforaminal epidural steroid injections.

### 2.5. Postoperative Utilization Analysis

During follow-up examinations, patients were interviewed regarding narcotic independence, the absence of incisional pain, and their functional status according to the modified Macnab criteria [73]. The authors intentionally did not further investigate the return-to-work dynamic as this problem was studied and published in another peer-reviewed article [29]. Patients in the control group and any of the three treatment arms were questioned explicitly whether further interventions, such as spinal injections or surgeries, were necessary to control the same or similar symptoms for which the surgical decompression was performed. In that event, they were censored for the Kaplan-Meier analysis [87] described below.

### 2.6. Correlative Surgical Outcome Analysis

The clinical outcome and postoperative utilization analysis were done using IBM SPSS Statistics software, Version 27.0, employing descriptive statistics (range, median, mean and standard deviation), two-way cross-tabulation (counts and percentages), two-tailed and paired t-test, and ANOVA. The Pearson χ^2^ and the likelihood-ratio χ^2^ tests were used as statistical measures of association. Kaplan-Meier (K-M) survival time probabilities and curves were constructed from tables containing (1) patients’ serial time and (2) their status at serial time (Macnab outcome—excellent, good, fair, or poor). Patients who dropped out of the study were lost to follow-up or in whom the required data was unavailable were censored [86]. Any patient with any additional surgery, regardless of whether at the index or another lumbar level, was censored. Control group patients, who, during their first encounter, declined any surgical or interventional care with spinal injections, were censored if they crossed over into interventional and surgical care plans. Patients with postoperative dysesthesia were not censored as this common sequela is not a complication requiring additional treatment beyond supportive care measures and spinal injections. The cumulative probability of having recovered, excluding censored events, is seen on the Y-axis of the K-M plot. The summed results for each group were added to derive the ultimate χ^2^ to compare the full K-M curves. The confidence intervals (95%) for the likelihood ratios were calculated using the log-rank method. The study’s IRB approval number is CEIFUS 106-19.

## 3. Results

Patients most frequently underwent surgery at L4/5 (141/345; 41.2%) followed by L5/S1 (83/345; 24.1%) and L4-S1 (59/345; 17.1%). The level distribution is shown in Table 1.

During the endoscopic operation, the most commonly visualized primary pain generator in the 206 ELD patients in descending order were hypertrophied ligamentum flavum (20.4%), contained herniated disc (15.0%), hypertrophied superior articular process (13.1%), inflamed disc with toxic annular tear (12.1%), extruded herniated disc (11.7%), delaminated and fissured disc tissue (8.3%), intra-annular granulation tissue (4.4%), facet cyst (3.9%), hidden shoulder osteophyte (3.4%), inflamed nerve (2.9%), tethered and furcal nerve roots (2.9%), and contracted foraminal ligaments (1.9%). In nearly half (46.6%; 96/206) of ELD patients, the MRI scan did not detect the painful pathology (Table 2).

Clinical function analysis at two years showed that 43.5% (150/345) of the surgically treated patients reported excellent Macnab outcomes, followed by 45.2% (156/345) with good, 9.3% (32/345) with fair, and 2.0% (7/345) with poor outcomes, respectively. A combined total of 88.7% of surgical patients reported their two-year postoperative functioning as excellent and good. Control group patients 23.9% (16/67) reported excellent Macnab outcomes, followed by 35.8% (24/67) with good, 28.4% (19/67) with fair, and 11.8% (8/67) with poor outcomes, respectively. Control patients crossed over into ELD (13.4%), TLIF (13.4%), laminectomy (10.4%) and interventional treatment (40.3%) arms at high rates. Most control patients treated with spinal injections (55.5%) had excellent and good functional outcomes versus 40.7% with fair and poor (3.7%). The majority of the 15 control patients (14/15; 93.4%) who remained on medical management without surgery or interventional care had the worst functional outcomes, rated as fair (46.7%) and poor (46.7%). ELD treated crossover control patients had 100% excellent Macnab outcomes at -two-year follow-up (9/9), versus 14.3% (1/1) excellent and 85.7% (6/7) good Macnab outcomes in control patients who crossed over into the laminectomy treatment arm. Control patients who crossed over into the TLIF arm had good two-year Macnab outcomes in 88.9% (8/9) and fair in 11.1% (1/9). The two-year outcomes according to Macnab criteria for the control group and surgical patients are listed in Table 3. 

The mean VAS score for leg pain reduced from preoperative 7.04 ± 1.85 to 3.45 ± 1.71 immediately postoperatively (*p* < 0.001), and 2.81 ± 1.69 at two-year follow-up (*p* < 0.001). Additional interventional and surgical treatments during the available follow-up period led to the censoring of 53.4% (220/412) of the study patients. However, no other treatment was required in 60.7% (125/206) of the ELD, 39.9% (31/78) of the TLIF, and 19.7% (12/61) of the laminectomy patients. In control patients, the most common symptom leading to cross-over and, thus, censoring was recurrent pain (76.1%; 51/67) and persistent pain (23.9%; 16/67). The most common new-onset symptom for censoring in ELD patients was recurrent pain (6.8%; 14/206), axial back pain in TLIF (25.6%; 20/78), and recurrent pain in laminectomy (65.6%; 40/61) patients (*p* < 0.001). The array of new symptoms that occurred after an initial period of functional improvement and prompted additional interventions and surgeries after the index operation by surgery type are listed in Table 4.

The types of additional interventions and surgeries that were done after the index operation and prompted censoring of the patient from the study are listed in Table 5. Transforaminal epidural steroid injections were tried in 29.7% (27/67) of control, 11.7% (24/206) of ELD, 23.1% (18/78) of TLIF, and 36.1% (22/61) of the laminectomy patients. The fusion rate among ELD patients was 8.8% (18/206). Among TLIF patients, the most common additional treatments were revision fusion (19.2%; 15/78) and multilevel rhizotomy (10.3%; 8/78). Common follow-up procedures in laminectomy patients included revision laminectomy (16.4%; 10/61), revision ELD (11.5%; 7/61), and multilevel rhizotomy (11.5%; 7/61). Transforaminal epidural steroid injections were tried in 11.7% (24/206) of ELD, 23.1% (18/78) of TLIF, and 36.1% (22/61) of laminectomy patients. The fusion rate among ELD patients was 8.8% (18/206). Among TLIF patients, the most common additional treatments were revision fusion (19.2%; 15/78) and multilevel rhizotomy (10.3%; 8/78). Typical follow-up procedures in laminectomy patients included revision laminectomy (16.4%; 10/61), revision ELD (11.5%; 7/61), and multilevel rhizotomy (11.5%; 7/61).

The Kaplan-Meier survival analysis showed that all study patients’ overall median durability of the lumbar decompression surgery was 38 months (Figure 5). In patients with excellent Macnab outcomes, the estimated median durability was 62 months in ELD, 43 in TLIF, and 31 months in laminectomy patients with excellent Macnab outcomes (*p* < 0.001; Figure 5, Figure 6, Figure 7 and Figure 8). In the control group patients, the median survival time was only seven months in patients with excellent and good outcomes because these patients crossed over to one of the surgical treatment arms. Control patients with fair Macnab outcomes were treated with spinal injections and had a median survival time of 6 months. Control patients with poor Macnab outcomes went largely untreated and had a mean survival time of 71 months until they were lost in follow-up.

In patients with Good Macnab outcomes, the estimated durability was similar regardless of the type of the index surgery: ELD = 31 months, TLIF = 31 months, and laminectomy = 29 months. The estimated median survival time for patients with fair Macnab outcomes was much less in minimally invasive TLIF (12 months) and open laminectomy (9 months) patients than in ELD patients (16 months), suggesting a greater severity of the postoperative symptoms in the TLIF and laminectomy patients compared to ELD patients. None of the patients with poor Macnab outcomes lasted very long after their failed index operation. The median survival times were six months or less regardless of whether they had an ELD (6 months), TLIF (5 months), or a laminectomy surgery (4). These differences were statistically significant when testing of equality of survival distributions for the different levels of the Macnab outcome variable was done with the Log Rank Mantel-Cox test (Chi-Square = 72.941.92, df = 3, *p* < 0.01).

The durability scenarios shown in Figure 5, Figure 6, Figure 7 and Figure 8 represent different natural history scenarios representing the clinical course after each of the three surgical interventions and the control group. The 74.1% (123/166) patients with excellent Macnab outcomes had no unplanned follow-up interventions or additional surgeries during the entire postoperative follow-up period.

## 4. Discussion

A personalized, minimally invasive, and targeted surgical lumbar decompression may effectively alleviate patients’ spine pain [88,89]. The surgical burden on the patient is significantly reduced with smaller incisions producing decreased perioperative surgical morbidity due to reduced blood loss, postoperative pain, earlier narcotic independence, and fewer complications [90]. One of the significant findings of our long-term study with greater than 5-year follow-up available in 80.58% (332/412) of study patients is that it is possible to reduce lumbar spine pain with a very small targeted transforaminal endoscopic decompression of the symptomatic neuroforamen in a similar fashion as observed with minimally invasive TLIF and traditional open laminectomy. This initial benefit overlap between the three studied lumbar surgery types was recently corroborated by an agnostic meta-analysis that investigated VAS-back and leg pain and Oswestry disability index improvements. [91] Normalized effect size analysis showed functional outcome improvements with the targeted ELD to be on par with TLIF and laminectomy surgeries compared to control group patients. The illustrative application of the SpineScreen protocol in our ELD patients suggests that predominant pain generators [92] can be identified using preoperative workup protocols with diagnostic injections. Our study also illustrates that overly relying on advanced image study reporting may leave patients without treatment since we were able to endoscopically directly visualize a primary pain generator in 46.6% of our 206 ELD study patients without a supporting MRI report (MRI false negative). The TLIF and laminectomy patients were only treated by employing image-based medical necessity criteria for surgery.

Moreover, our study confirms that it is possible to achieve high patient satisfaction with a statistically significant reduction in VAS scoring for leg pain and functional improvements in the long run by deliberately ignoring other potential pain generators suggested by the preoperative MRI scan and its report. Hence, using SpineScreen criteria rather than image-based medical necessity criteria for lumbar spine surgery for stratifying our TLIF and laminectomy patients produces no worse initial two-year clinical outcomes. These observations of our study have also been corroborated by several comparative randomized prospective clinical outcomes studies comparing ELD to microsurgical and open decompression for unrelenting lumbar spine pain [93,94,95]. The differences in the clinical value proposition of the SpineScreen methodology employing diagnostic injections before the ELD procedure become apparent when analyzing the postoperative natural history with the Kaplan-Meier durability analysis of each of the three lumbar surgeries in comparison to a control group of patients who initially declined the surgical recommendation and went on to continue with medical management and physical therapy.

For several decades, this article’s first and senior authors have applied this context-driven spine care model in their routine clinical practice. [26,28,48,51,52,96] The authors’ methodology (SpineScreen) is similar to that of the Society for Brain Mapping and Therapeutics (SBMT), which proposed a multifaceted approach to Brain, Spine, and Mental Health Screening (NeuroScreen). [97] It will be described in the upcoming SBMT position paper on modern management of painful spine conditions. The illustrative case series presented in this article highlights the efficacy of the authors’ methodology by graphically demonstrating the duration of symptom relief in the Kaplan-Meier curves with a longer overall estimated median survival time of 54 months after ELD surgery compared to minimally invasive TLIF (36 months) and open laminectomy (29 months). As illustrated in Table 6, patients with Excellent Macnab outcomes had much longer survival times at a statistical significance level of *p* < 0.001 with estimated median durability of 62 months compared to 43 months in TLIF and 31 months in laminectomy patients (Figure 5, Figure 6, Figure 7 and Figure 8).

Further, this article has its foundation in a set of peer-reviewed articles published by the authors, demonstrating a comparatively low fusion rate of 8.9% [88] and 2.3% [89] in their endoscopic spine patients. These numbers are much lower when compared to the numbers reported with traditional translaminar open spine surgery. The significance of such low 5-year fusion conversion rates regarding health care cost savings is tremendous. One open laminectomy study including 5,636 patients over 60 years reported an overall 5-year revision rate of 16.5%, with mechanical failure being the main reason for a secondary revision operation. [98] Considering the high number of spinal stenosis decompression, with about 600,000 surgeries performed in the United States annually, [98] hospital admissions and inpatient surgery costs related to lumbar spinal stenosis in patients over 65 have risen substantially and were estimated to be $1.65 Billion in 2007. [99] The rate of complex fusion procedures increased 15-fold, from 1.3 to 19.9 per 100,000 Medicare beneficiaries from 2002 to 2007. [100] The same study reported an increase in life-threatening complications from 2.3% in patients who underwent decompression to 5.6% in those who had complex fusions with higher thirty-day rehospitalization rates in fusion patients - 0.8% in decompression versus 13.0% in complex fusion - resulting in higher adjusted mean hospital charges for complex fusion surgeries of $80,888 compared with $23,724 for decompression alone. [101] Lumbar fusion spending increased more than 500%, from 75 million dollars to 482 million dollars from 1992, when it represented 14% of total spending for back surgery, to 47% by 2003. [102] As the cost of modern spine care is rising [31,33,42,103,104,105] spine surgery may not be affordable unless rationed. Current advances have only added to the affordability crisis. Our long-term study shows that performing an index operation with a low mechanical failure rate due to iatrogenic instability is critical to controlling costs in spine surgery.

Over the last 40 years, open traditional spine surgery has established a track record that is well accepted as a standard for comparison [105]. One recent 10-year study illustrated that the revision surgery rate for spinal fusion to treat degenerative spine disease from adjacent segment disease alone was 20.4%. [106] Additional indications for revision spine surgery do exist as well. While in patients suffering from symptomatic osteoarthritis of the hip or knee, the definitive treatment can be rolled into one treatment episode by joint replacement surgeries [107], this approach is not practical in spine surgery. Multilevel spinal segment involvement is common. It would likely lead to overutilization by performing surgery on non-symptomatic levels whose MRI appearance is abnormal. [108,109] This dynamic creates a large group of untreated patients considered too young or too old for surgery because they do not meet the image-based medical necessity criteria for lumbar surgery. One study found the group of false-negative MRI reads to be as high as 35%. [1,110] Others are escaping definitive spine care because they have too many co-morbidities. Increased scrutiny on spine surgery by patients and the stakeholders on the payers’ side created the need for more simplified innovative, less costly, and more effective treatments with good durability.

The latter is undoubtedly supported by the findings of the authors’ study. The authors previously used the Kaplan-Meier survival analysis to establish the superiority of endoscopic over laser decompression. [111] This study is the first to formally analyze the long-term durability of the treatment benefit with the targeted transforaminal endoscopic decompression employing Kaplan-Meier survival analysis. It confirms the feasibility of using a less invasive transforaminal approach in a more personalized spine care model. Such models may also benefit from earlier intervention to allow the disease to heal and prevent it from running to its end-stage, where aggressive and costly fusions are often the only effective salvage operation. Its results showed that the most common new-onset symptom for censoring was recurrent pain in ELD (6.8%; 14/206), axial back pain in TLIF (25.6%; 20/78), and recurrent pain in laminectomy (65.6%; 40/61) patients (*p* < 0.001). Transforaminal epidural steroid injections were tried in 11.7% (24/206) of ELD, 23.1% (18/78) of TLIF, and 36.1% (22/61) of laminectomy patients. The fusion rate among ELD patients was 8.8% (18/206). Among TLIF patients, the most common additional treatments were revision fusion (19.2%; 15/78) and multilevel rhizotomy (10.3%; 8/78). Common follow-up procedures in laminectomy patients included revision laminectomy (16.4%; 10/61), revision ELD (11.5%; 7/61), and multilevel rhizotomy (11.5%; 7/61). The remainder of the additional procedures were performed on patients who were censored well after the two-year follow-up examination suggesting that their additional minimally invasive targeted ablation- and decompression procedures were prompted by the natural progression of the underlying degenerative spine disease at adjacent or distant levels and not by the index operation.

Our study’s Kaplan-Meier survival curves illustrate the durability of the treatment benefit that can be achieved with comprehensive, personalized spine care via targeted decompression procedures versus the more traditional image-based TLIF and laminectomy surgeries. In addition to the retrospective nature of our study, the most significant limitation is that the Kaplan-Meier curves are not a prognosis of the outcome. The accuracy of the Kaplan-Meier curves deteriorates as soon as the first patient is censored from the study. However, they effectively illustrate the various postoperative clinical dynamics that played out in the different outcome groups due to the personalized lumbar endoscopic decompression done in an ambulatory surgery center as opposed to with TLIF and laminectomy done at a hospital while using different necessity criteria for surgery. Others [112,113,114] demonstrated that not performing these decompressions in an inpatient setting reduces cost. Endoscopic spine procedures can meet the increasing demand for these simplified surgical treatments. Our study suggests that ELD surgery on patients stratified using SpineScreen provides spine care that is not only effective initially but also may survive better in the long run with far fewer follow-up interventions and reoperations. In addition, our study shows that personalized spine care minimizes the need for aggressive reoperations. Future investigations should determine whether it can reduce the repetition of ineffective treatment options currently in routine clinical use in non-surgical population management-oriented spine care. The latter care model weeds out patients to fit the image-based necessity criteria for more traditional spine surgery. These stipulations seem more relevant now than ever in the current environment where physicians face lower reimbursement, spend more time on uncompensated tasks, and have to cope with higher overhead, higher technology costs, and lost revenue or increased expenses due to the COVID-19 pandemic.

The SpineScreen methodology should be validated in future studies, preferably in the more scrutinized and controlled setting typical of prospective, randomized, or at a minimum propensity score-adjusted multicenter studies with a larger sample size and more extended follow-up period to confirm our results. However, randomized trials in surgical settings bring their challenges, as McCulloch and colleagues [115] pointed out 20 years ago:Structural, cultural, and psychological resistance to randomization,The inherent variability of surgical procedures and the need for a precise definition of interventions and quality monitoring,Surgical learning curves of new techniques,Patient’s equipoise, given the differences in the risks of surgical and non-surgical treatments,Difficulties with recruitment.Consent and randomization in rare conditions and urgent and life-threatening situations.

Factors identified by others [116,117,118,119] include the difficulty of blinding patients and assessors; the effects of clustering by surgeon or center; standardization of the surgical intervention; and, on the patient side, expectation bias, especially if surgery was preceded by non-surgical interventions as well as placebo and nocebo effects. As Solheim points out, surgical trials “may reach a glass ceiling in the climb up the hierarchy of evidence,” implying that the rigor of drug trials and the associated strength of evidence may not be reachable on a large scale in surgical trials.

## 5. Conclusions

SpineScreen describes a multifaceted methodology that aims to provide personalized, targeted spine care. This care model is an alternative to traditional image-based necessity criteria for spinal surgery. The degenerative spine disease is often poorly managed in its initial stages and only treated definitively in its end stages, where reconstructive decompression fusions are often the only reasonable option. As routinely done in managing other chronic diseases, the authors propose using their staged management style for earlier intervention in painful spine disease to treat the large group of patients that are currently unattended yet seek help from other physicians providing chiropractic homeopathic and pain management care. The high utilization in these other subspecialties and the high cost of end-stage traditional surgical spine care associated with a higher reoperation rate warrant additional research to determine whether the authors’ personalized spine care model focused on a more targeted approach to the patients’ predominant pain generator can be applied to the primary care setting or to larger referral centers and specialty clinics where follow through on the complex SpineScreen protocol may be less practical due to high staff turnover or variations in surgeon skill level. At a minimum, the authors demonstrated that different postoperative performance characteristics between the investigated lumbar surgeries exist, and they may impact the outcome and indirectly cost.

## Figures and Tables

**Figure 1 jpm-12-01065-f001:**
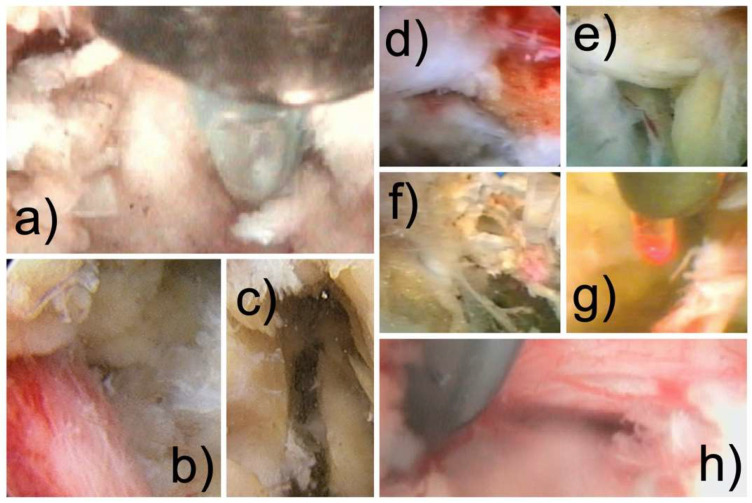
Examples of directly visualized pain generators identified with the spinal endoscope that are often missed by routine lumbar MRI scans include herniated or inflamed disc herniation (**a**), an inflamed nerve (**b**), a superior foraminal facet osteophyte, a vacuum disc with delaminated devitalized disc tissue (**c**), a hypertrophied superior articular process (SAP) with a facet cysts and tender capsule, an impacting facet margin (**d**), a furcal nerve (**e**), a tethered nerve root with contracted scar tissue from the pars inter-articularis (**f**), a hypervascular scar on the exiting nerve root (**g**), and a hypervascular in-flamed traversing nerve root (**h**).

**Figure 2 jpm-12-01065-f002:**
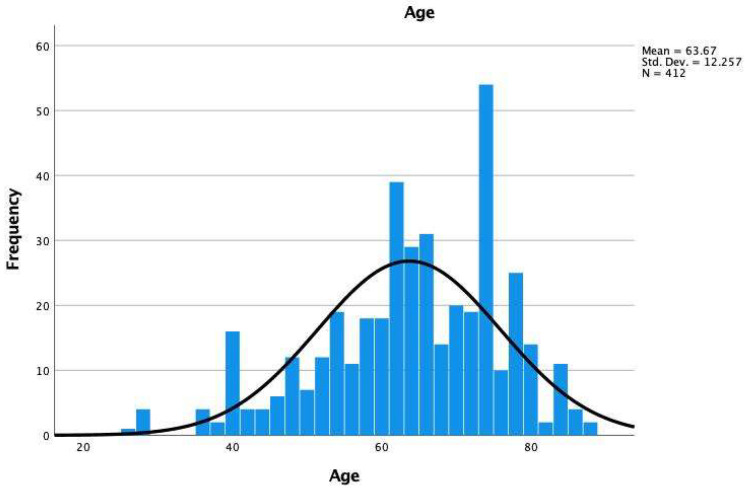
Histogram of the age distribution of the 412 study patients with a mean age of 63.67 ranging from 26 to 88 years with a standard deviation of 12.257 years.

**Figure 3 jpm-12-01065-f003:**
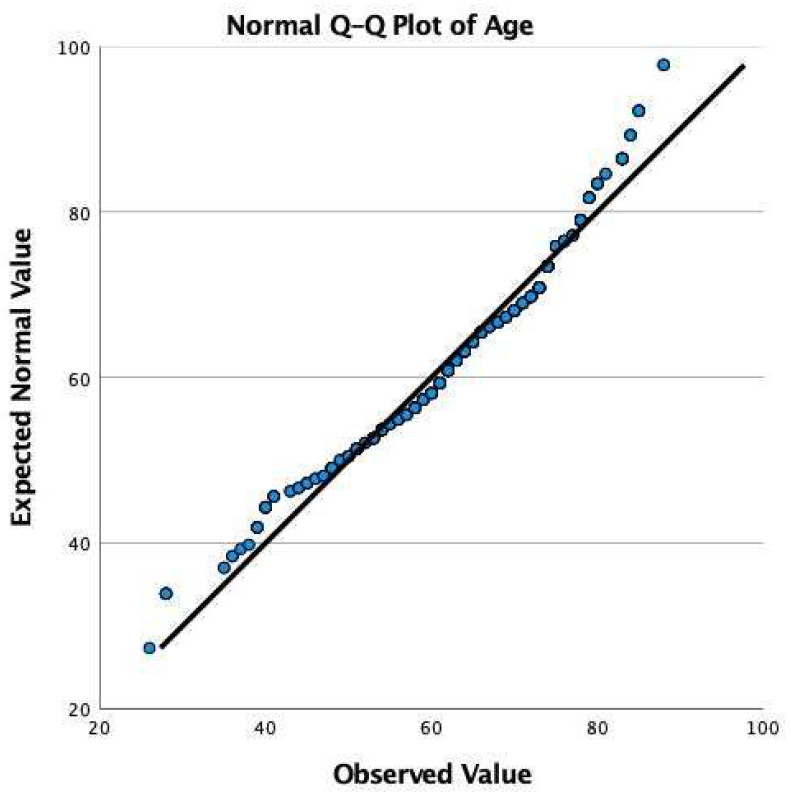
Q-Q Plot of expected versus observed values of age confirming the normal distribution of the 412 study patients’ age.

**Figure 4 jpm-12-01065-f004:**
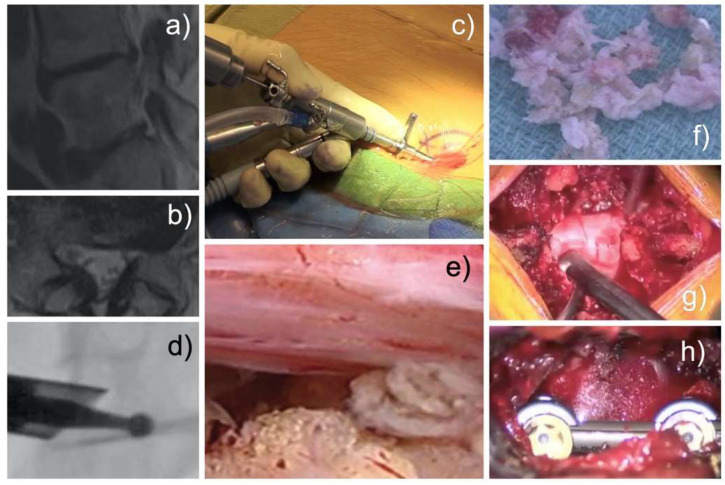
Preoperative T2-weighted sagittal (**a**) and axial MRI scans of a typical patient with multi-level lumbar degenerative disease and symptomatic L4/5 foraminal stenosis are shown (**a**,**b**). The patient underwent directly visualized ELD (**c**), with drills (**d**) and rongeurs introduced through a spinal endoscope’s inner working channel (**e**). In this case, the primary pain generator (PPG) was extruded disc fragments removed piecemeal after performing a foraminoplasty (**f**). Examples of an open laminectomy wound are shown in panel (**g**), and minimally invasive TLIF incision using a 22 mm tubular retractor are shown in panel (**h**).

**Figure 5 jpm-12-01065-f005:**
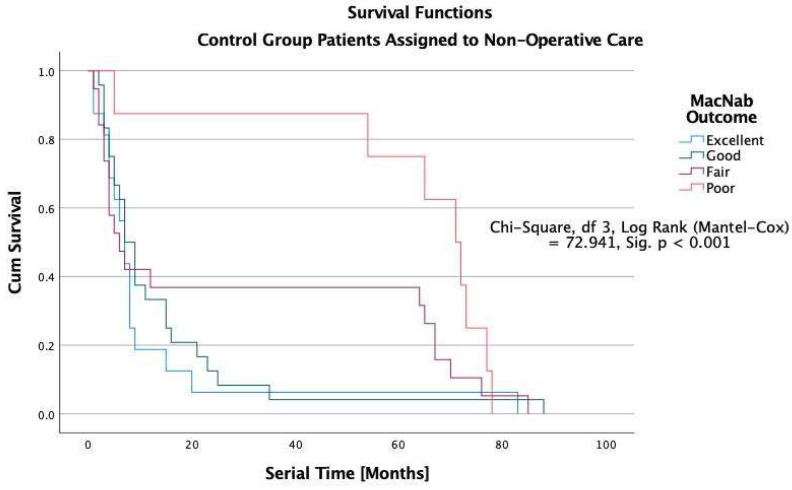
The estimated median (50% percentile) survival times for each of the three surgery types by Macnab outcome criteria are listed in Table 4. The estimated median (50% percentile) durability (survival) among all 412 study patients was 38 months with a standard error of 1.864, a lower boundary of 34.347, and an upper boundary of 41.653. Patients were censored if they required additional treatment after their index operation or if their outcome was not known at their final follow-up. Control patients were censored if they failed conservative care and crossed over into one of the surgical treatment groups or required interventional pain management care with spinal injections. Control group patients were recruited from patients who, during their first encounter, declined any surgical or interventional care with spinal injections and were treated with medical pain management and active physical rehabilitation programs. Of the control patients, the crossover rate into interventional and surgical care was 40.3% and 37.2%, respectively. Only 15 of the 67 (22.4%) control patients continued with conservative care until final follow-up; all of which had fair and poor functional Macnab outcomes, thus, explaining the short overall survival time in control patients of only eight months with a standard error of 0.942, a lower boundary of 6.154 and an upper boundary of 9.846 months.

**Figure 6 jpm-12-01065-f006:**
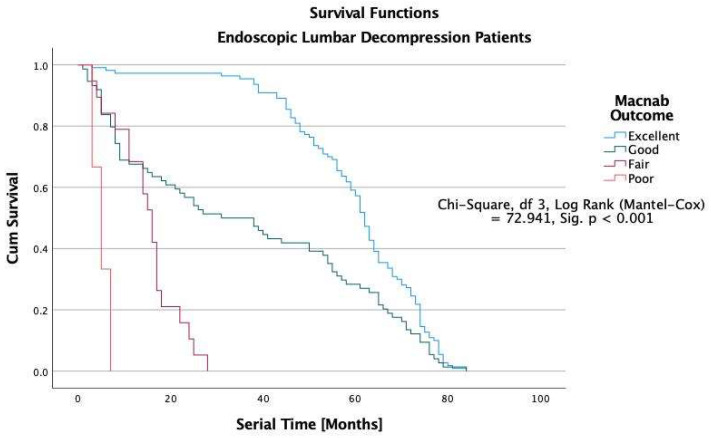
Kaplan-Meier (K-M) Survival functions by Macnab outcomes graphically illustrate the durability of ELD surgery. Patients were censored if they required additional treatment after their index operation or if their outcome was not known at their final follow-up. The estimated median (50% percentile) overall durability (survival) among ELD patients (*n* = 206) was 54 months with a standard error of 2.392, a lower boundary of 49.311, and an upper boundary of 58.689 months. The survival time in ELD patients with Excellent Macnab outcomes was 62 months with a standard error of 0.982, a lower boundary of 60.076, and an upper boundary of 63.924 months (Table 4).

**Figure 7 jpm-12-01065-f007:**
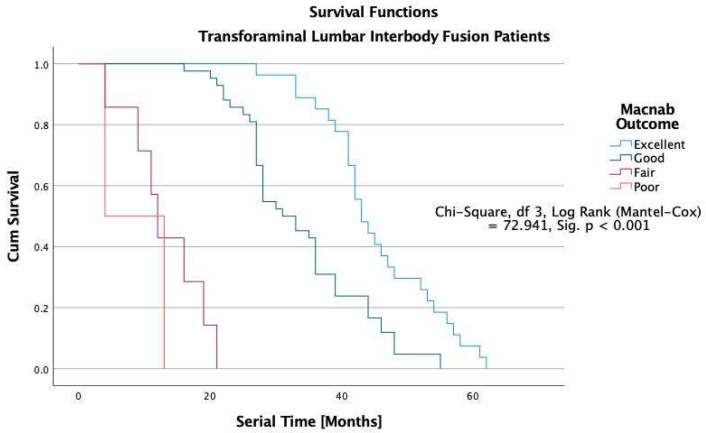
Kaplan-Meier (K-M) Survival functions by Macnab outcomes graphically illustrate the durability of laminectomy surgery. Patients were censored if they required additional treatment after their index operation or if their outcome was not known at their final follow-up. The estimated median (50% percentile) overall durability (survival) among laminectomy patients (*n* = 61) was 29 months with a standard error of 1.672, a lower boundary of 25.724, and an upper boundary of 32.276 months. The estimated median survival time in laminectomy patients with Excellent Macnab outcomes was 31 months with a standard error of 3.595, a lower boundary of 23.954, and an upper boundary of 38.046 months (Table 4).

**Figure 8 jpm-12-01065-f008:**
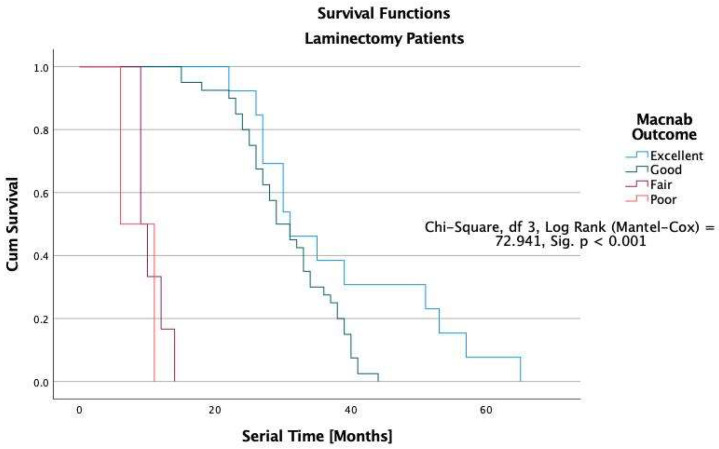
Kaplan-Meier (K-M) Survival functions by Macnab outcomes graphically illustrate the durability of TLIF surgery. Patients were censored if they required additional treatment after their index operation or if their outcome was not known at their final follow-up. The estimated median (50% percentile) overall durability (survival) among TLIF patients (*n* = 78) was 36 months with a standard error of 2.752, a lower boundary of 30.607, and an upper boundary of 41.393 months. The survival time in TLIF patients with Excellent Macnab outcomes was 43 months with a standard error of 1.731, a lower boundary of 39.608, and an upper boundary of 46.392 months (Table 4).

**Table 1 jpm-12-01065-t001:** Level distribution of surgical levels of patients treated with ELD, Laminectomy, and TLIF.

Sugical Level	Frequency	Percent	Cumulative Percent
L2/3	18	4.4	4.4
L2/3 L3/4	2	0.5	4.9
L3-5	4	1.0	5.8
L3/4	28	6.8	12.6
L3/4 L5/S1	2	0.5	13.1
L3/4 L4-S1	6	1.5	14.6
L3/4 L4/5	12	2.9	17.5
L4-S1	67	16.3	33.7
L4/5	170	41.3	75.0
L5/S1	103	25.0	100.0
Total	412	100.0	

**Table 2 jpm-12-01065-t002:** Crosstabulation primary pain generator visualized during the endoscopic operation and their reporting on routine lumbar MRI.

Endoscopically Visualized Pain Generator	MRI Negative	MRI Positive	Total:
Hypertrophied Ligamentum Flavum	7	35	42
	7.3%	31.8%	20.4%
Contained Herniated Disc	6	25	31
	6.3%	22.7%	15.0%
Hypertrophied Superior Articular Process	3	24	27
	3.1%	21.8%	13.1%
Inflammed Disc With Toxic Annular Tear	25	0	25
	26.0%	0.0%	12.1%
Extruded Herniated Disc	5	19	24
	5.2%	17.3%	11.7%
Delaminated And Fissured Disc Tissue	17	0	17
	17.7%	0.0%	8.3%
Intra-Annular Granulation Tissue	9	0	9
	9.4%	0.0%	4.4%
Facet Cyst	1	7	8
	1.0%	6.4%	3.9%
Hidden Shoulder Osteophyte	7	0	7
	7.3%	0.0%	3.4%
Inflamed Nerve	6	0	6
	6.3%	0.0%	2.9%
Tethered and Furcal Nerve Roots	6	0	6
	6.3%	0.0%	2.9%
Contracted Foraminal Ligaments	4	0	4
	4.2%	0.0%	1.9%
Total ELD Patients	96	110	206
	100.0%	100.0%	100.0%

MRI Negative: The radiologist did not describe the endoscopically visualized primary pain generator in the MRI report. MRI Positive: The radiologist did describe the endoscopically visualized primary pain generator in the MRI report.

**Table 3 jpm-12-01065-t003:** Macnab Outcomes with ELD, TLIF, and Laminectomy.

	Control	Index Surgery Type			Total
Macnab Outcome	Recruited from patients referred for surgery, who declined	ELD	TLIF	Laminectomy	
Excellent	16	110	27	13	166
	9.6%	66.3%	16.3%	7.8%	100.0%
Good	24	74	42	40	180
	13.3%	41.1%	23.3%	22.2%	100.0%
Fair	19	19	7	6	51
	37.3%	37.3%	13.7%	11.8%	100.0%
Poor	8	3	2	2	15
	53.3%	20.0%	13.3%	13.3%	100.0%
Total:	67	206	78	61	412
	16.3%	50.0%	18.9%	14.8%	100.0%

**Table 4 jpm-12-01065-t004:** New onset postoperative symptoms prompting new treatments after the index operation.

New Onset Postoperative Symptom			Type of Lumbar Index Surgery			Total
		Control	ELD	Laminectomy	TLIF	
N/A		0	143	37	12	192
		0.0%	74.4%	19.3%	6.3%	100.0%
Axial back pain		0	11	20	9	40
		0.0%	27.5%	50.0%	22.5%	100.0%
Other level pain		0	12	4	0	16
		0.0%	75.0%	25.0%	0.0%	100.0%
Persistent Pain		16	3	2	0	21
		76.2%	14.3%	9.5%	0.0%	100.0%
Recurrent HNP		0	9	0	0	9
		0.0%	100.0%	0.0%	0.0%	100.0%
Recurrent pain		51	14	6	40	111
		45.9%	12.6%	5.4%	36.0%	100.0%
Sacral Iliac Joint Pain		0	1	0	0	1
		0.0%	100.0%	0.0%	0.0%	100.0%
Same level other side pain		0	11	9	0	20
		0.0%	55.0%	45.0%	0.0%	100.0%
Total:		67	206	78	61	412
		16.3%	50.0%	18.9%	14.8%	100.0%
Chi-Square Tests			df	Asymptotic Significance (2-sided)		
Pearson Chi-Square = 312.275			24	*p* < 0.001		
Likelihood Ratio = 339.800			24	*p* < 0.001		
	N of Valid Cases: 412					

ELD—Endoscopic Lumbar Decompression; TLIF—Transforaminal Lumbar Interbody Fusion.

**Table 5 jpm-12-01065-t005:** Additional treatments for persistent or new-onset symptoms in control group patients and ELD patients following the targeted lumbar endoscopic decompression of visualized painful pathology based on SpineScreen, versus MRI-based laminectomy and TLIF.

Postoperative Treatments		Surgery Type			Total
	Control	ELD	TLIF	Laminectomy	
N/A	15	125	31	12	183
	8.2%	68.3%	16.9%	6.6%	100.0%
TESI	27	24	18	22	91
	29.7%	26.4%	19.8%	24.2%	100.0%
Adjacent Level TLIF	0	0	2	0	2
	0.0%	0.0%	100.0%	0.0%	100.0%
ASD Fusion	0	0	3	0	3
	0.0%	0.0%	100.0%	0.0%	100.0%
ASD Laminectomy	0	0	1	0	1
	0.0%	0.0%	100.0%	0.0%	100.0%
ELD	9	0	0	0	9
	100.0%	0.0%	0.0%	0.0%	100.0%
ELD Adjacent Level	0	11	0	0	11
	0.0%	100.0%	0.0%	0.0%	100.0%
ELD Opposite side	0	11	0	0	11
	0.0%	100.0%	0.0%	0.0%	100.0%
ELD same side and level	0	2	0	0	2
	0.0%	100.0%	0.0%	0.0%	100.0%
Hemilaminectomy	0	1	0	0	1
	0.0%	100.0%	0.0%	0.0%	100.0%
Laminectomy	7	0	0	0	7
	100.0%	0.0%	0.0%	0.0%	100.0%
Multilevel Laminectomy	0	1	0	0	1
	0.0%	100.0%	0.0%	0.0%	100.0%
Multilevel Rhizotomy	0	7	8	7	22
	0.0%	31.8%	36.4%	31.8%	100.0%
Repeat ELD For Recurrent HNP	0	2	0	0	2
	0.0%	100.0%	0.0%	0.0%	100.0%
Revision ELD	0	0	0	7	7
	0.0%	0.0%	0.0%	100.0%	100.0%
Revision Laminectomy	0	0	0	10	10
	0.0%	0.0%	0.0%	100.0%	100.0%
Revision TLIF	0	14	15	3	32
	0.0%	43.8%	46.9%	9.4%	100.0%
Same and Adjacent Level TLIF	0	1	0	0	1
	0.0%	100.0%	0.0%	0.0%	100.0%
Same Level ALIF	0	3	0	0	3
	0.0%	100.0%	0.0%	0.0%	100.0%
Same Level Laminectomy	0	1	0	0	1
	0.0%	100.0%	0.0%	0.0%	100.0%
Same Level Rhizotomy	0	2	0	0	2
	0.0%	100.0%	0.0%	0.0%	100.0%
SI Ablation	0	1	0	0	1
	0.0%	100.0%	0.0%	0.0%	100.0%
TLIF	9	0	0	0	9
	100.0%	0.0%	0.0%	0.0%	100.0%
	67	206	78	61	412
	16.3%	50.0%	18.9%	14.8%	100.0%
Chi-Square Tests	df		Asymptotic Significance (2-sided		
Pearson Chi-Square = 374.425	66		*p* < 0.001		
Likelihood Ratio = 317.104	66		*p* < 0.001		
N of Valid Cases: 412					

**Table 6 jpm-12-01065-t006:** Medians for survival time by Macnab outcome estimating the durability of the targeted lumbar decompression of visualized painful pathology based on SpineScreen (ELD) versus MRI-based laminectomy and TLIF.

Group	Macnab Outcome	Median			
		Estimate	Std. Error	95% Confidence Interval	
				Lower Bound	Upper Bound
Control	Excellent	7.000	0.992	5.055	8.945
	Good	7.000	1.225	4.600	9.400
	Fair	6.000	2.176	1.734	10.266
	Poor	71.000	4.950	61.298	80.702
	Overall	8.000	0.942	6.154	9.846
ELD	Excellent	62.000	0.982	60.076	63.924
	Good	31.000	8.124	15.076	46.924
	Fair	16.000	1.435	13.188	18.812
	Poor	5.000	1.633	1.799	8.201
	Overall	54.000	2.392	49.311	58.689
Laminectomy	Excellent	31.000	3.595	23.954	38.046
	Good	29.000	1.897	25.281	32.719
	Fair	9.000	.	.	.
	Poor	6.000	.	.	.
	Overall	29.000	1.672	25.724	32.276
TLIF	Excellent	43.000	1.731	39.608	46.392
	Good	31.000	2.592	25.919	36.081
	Fair	12.000	1.309	9.434	14.566
	Poor	4.000	.	.	.
	Overall	36.000	2.752	30.607	41.393
Overall	Overall	38.000	1.864	34.347	41.653

ELD—Endoscopic Lumbar Decompression; TLIF—Transforaminal Lumbar Interbody Fusion.

## Data Availability

The data presented in this study are available on request from the corresponding author.

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
