# Peer review of "A Proposed Personalized Spine Care Protocol (SpineScreen) to Treat Visualized Pain Generators: An Illustrative Study Comparing Clinical Outcomes and Postoperative Reoperations between Targeted Endoscopic Lumbar Decompression Surgery, Minimally Invasive TLIF and Open Laminectomy"

_jpm, 2022, doi:10.3390/jpm12071065_

Round 1

Reviewer 1 Report

The work was done well. The methodology has been described in detail and the results are clear.

Author Response

We appreciate this reviewers comments.

Reviewer 2 Report

I appreciated the paper by Lewandrowski and colleagues on the use of SpineScreen to treat visualized pain generators. The necessity to apply targeted medicine is nowadays mandatory, and finding new protocols are necessary. The structure of the paper is well written and the methodology is valid. 

However the english is not fluent and the article is very difficult to read. I suggest an extensive english revision.

Moreover the self citation rate is too high! I think also other authors wrote about this topic...

Tables 1-4 could be improved with a statistical comparison.

No limitations are reported in the discussion section

Author Response

We appreciate this reviewer's comments on the need for targeted medicine. This reviewer found the paper well written, and the methodology is valid. 
We had the paper copy-edited by a professional to improve English, fluency, and readability. I hope this reviewer finds the article more acceptable in its current form.
This article is co-authored by KOLs with a combined experience with the SpineScreen protocol of more than 45 years. We added more quotations to dilute the self-citation rate. Some self-citation is inevitable to present the prior art accurately. We hope the reviewer finds the addition of some 20 references explaining the high revision rate with traditional spine surgery and the limitations in a surgical clinical study as an acceptable solution.
We simplified many tables by eliminating mean data reporting and only reporting on mean survival data. We added another table on the frequencies of the endoscopically visualized pathologies that escaped the routine lumbar MRI scan to explain better the dilemma many spine surgeons find themselves in when making treatment recommendations to patients when the MRI fails to identify the problem.
We added a significant portion of our study's limitations. Specifically, the selection bias limitation of our observational study, the ceiling effect limitation of surgical clinical studies where randomization is often impractical, and the limitations of the Kaplan Meyer analysis, which is not a prediction of surgical outcome but merely an illustration of the postoperative dynamic of the various treatment arms. We hope the reviewer finds these explanations acceptable.